# Rational Metareasoning for Large Language Models

## Abstract

Being prompted to engage in reasoning has emerged as a core technique for using large language models (LLMs), deploying additional inference-time compute to improve task performance. However, as LLMs increase in both size and adoption, inference costs are correspondingly becoming increasingly burdensome. How, then, might we optimize reasoning's cost-performance tradeoff? This work introduces a novel approach based on computational models of metareasoning used in cognitive science, training LLMs to selectively use intermediate reasoning steps only when necessary. We first develop a reward function that incorporates the Value of Computation by penalizing unnecessary reasoning, then use this reward function with Expert Iteration to train the LLM. Compared to few-shot chain-of-thought prompting and STaR, our method significantly reduces inference costs (20-37% fewer tokens generated across three models) while maintaining task performance across diverse datasets.

## 1 Introduction

Large language models (LLMs) rely on substantial computational power to handle complex problems (OpenAI et al., 2024; Chowdhery et al., 2022; de Vries, 2023). While initial studies mostly focused on the cost of training (Verdecchia et al., 2023), LLMs' widespread deployment has made inference-time costs an increasingly important factor. Model compression techniques such as quantization, pruning, and knowledge distillation can lower post-training costs (Wan et al., 2024). However, there is a fundamental tension between inference cost and task performance: while many of these methods reduce costs at the expense of performance, others, such as chain-of-thought prompting (CoT; Wei et al., 2023; Kojima et al., 2023), do the opposite, raising inference costs to enhance task performance (Snell et al., 2024). It is worth noting that none of the previous approaches are *adaptive*: model compression modifications and existing CoT methods tend to raise or lower the inference cost on *all* queries, regardless of task complexity.

In stark contrast to this static tradeoff, humans are able to adaptively allocate computational resources based on task difficulty (Krämer, 2014; Russell, 1997; Lieder & Griffiths, 2017). In this work, we draw inspiration from *rational metareasoning* – literally, reasoning about reasoning – a concept originally from the artificial intelligence literature (Russell & Wefald, 1991) that has been used to explain how humans adaptively manage computational resources (Lieder & Griffiths, 2017; Lieder et al., 2018; Griffiths et al., 2019a). Building on this, we develop a novel reward function based on the Value of Computation (VOC; Russell & Wefald, 1991), which formalizes the trade-off between inference cost and task performance. We adopt an iterative reinforcement learning process inspired by the Expert Iteration algorithm (Anthony et al., 2017). In each iteration, we generate multiple reasoning chains for each question. These reasoning chains are ranked using the reward function, and the dataset is filtered to retain only the best reasoning chain for each question. The model is then fine-tuned using this filtered dataset. However, unlike previous applications of Expert Iteration

to LLMs (Zelikman et al., 2022), which filter generated examples based on correctness, our method optimizes for both the correctness *and the cost* of the reasoning process.

We evaluated the effectiveness of our solution across a diverse set of tasks, from science knowledge (ARC; Clark et al., 2018) to commonsense reasoning (CommonsenseQA; Talmor et al., 2019), mathematical problem solving (GSM8K; Cobbe et al., 2021), and logical deductive reasoning (ProofWriter; Tafjord et al., 2021). Additionally, we assess the out-of-domain generalization on MMLU (Hendrycks et al., 2021), a multitask benchmark. Our approach achieves a substantial reduction in generated tokens (35.6-37.4% compared to few-shot prompting and 20.1-27.3% compared to STaR; Zelikman et al. 2022) while retaining comparable performance. Thus, we make the following contributions:

1. We introduce the idea of using rational metareasoning to optimize the tradeoff between inference cost and performance of LLMs.

2. We formalize a novel reward function inspired by the Value of Computation (VOC) and integrate it into LLM training.

3. We empirically demonstrate that rational metareasoning achieves the same task performance at lower inference costs (20-37% fewer tokens on average) across various datasets and reasoning tasks.

## 2 RATIONAL METAREASONING

Unlike artificial intelligence, humans have limited time and cognitive resources (Griffiths et al., 2019b; Griffiths, 2020). We face diverse challenges requiring different approaches: avoiding a sudden obstacle when driving needs quick, intuitive thinking, whereas selecting a retirement investment strategy requires slow, deliberate reasoning (Krämer, 2014). Rational metareasoning (Russell & Wefald, 1991) suggests agents should adapt their reasoning based on the problem at hand. Intuitively, while reasoning solves a problem, metareasoning solves the problem of *how* to solve a problem: deciding which computations to perform while problem-solving. The essence of rational metareasoning is calculating the value of computation (VOC; Russell & Wefald, 1991) for each potential computation. The VOC balances the benefit of computation $c$ (characterized by the expected increase in the agent's eventual utility) against its cost (usually time or energy).

To formalize this, agents are assumed to have some internal belief state $b \in \mathcal{B}$, which determines their expectation about the value of each action $a \in \mathcal{A}$: $\mathbb{E}[U(a)|b]$. A rational agent would simply choose the highest-value action: $a^* = \arg\max_{a \in \mathcal{A}} [U(a)|b]$. In contrast, a meta-rational agent can perform computation to change their belief state before choosing an action. Each computation $c \in \mathcal{C}$ updates the agent belief to $b'$ with probability $P(b'|c)$, which in turn affects their beliefs about the value of actions. However, each computation has an associated cost (cost($c$)). The VOC then quantifies the value of performing computation $c$ given a starting belief state $b$,

$$VOC(c, b) = \mathbb{E}_{P(b'|c)}[\max_{a'} \mathbb{E}[U(a')|b'] - \max_{a} \mathbb{E}[U(a)|b]] - \text{cost}(c). \quad (1)$$

Thus, a meta-rational agent should pursue the computation $c^*$ with the highest VOC: $c^* = \arg\max_{c \in \mathcal{C}} VOC(c, b)$. If no computation has positive VOC, the agent should stop thinking and act in the world. Rational meta-reasoning can explain how humans allocate cognitive resources in various tasks (Lieder & Griffiths, 2017; Lieder et al., 2018; Callaway et al., 2018; 2021; 2022; Russek et al., 2022).

# 3 RATIONAL METAREASONING WITH LARGE LANGUAGE MODELS

To achieve an optimal balance between performance and efficiency, our approach introduces a new VOC-inspired reward function (Eq. 2) into an Expert Iteration training loop (Anthony et al., 2017; Zelikman et al., 2022), fine-tuning a LLM to produce reasoning chains adaptively depending on task difficulty.

## 3.1 REWARD MODELING

Chain-of-thought prompting is a technique in which a language model is encouraged to generate an intermediate output – a "chain of thought" – prior to producing the answer to a question (Wei et al., 2023; Kojima et al., 2023) We define the reward of a chain of thought as the difference between its utility and its cost,

$$\mathcal{R}_\pi(x, y, z) = \mathcal{U}_\pi(z|x, y) - \mathcal{C}(z) \tag{2}$$

where $x$ denotes the input for the task, $z$ represents the chain of thought, and $y$ is the target solution. The utility of the chain of thought is represented by $\mathcal{U}_\pi(z|x, y)$, and the cost of the intermediate computations is denoted by $\mathcal{C}(z)$. Equation 2 mirrors the VOC Equation 1: here, individual reasoning tokens correspond to intermediate computations $c$, making the reasoning chain $z$ a sequence of computations $c$, while the actions $a \in \mathcal{A}$ map to the potential outputs $y \in \mathcal{Y}$ of the language model. In the context of LLMs, utility quantifies the increase in the likelihood of generating the target sequence $y$ when the chain of thought $z$ is added to the input $x$, under the policy $\pi$:

$$\mathcal{U}_\pi(z|x, y) = \log \pi_\theta(y|z, x) - \log \pi_\theta(y|x). \tag{3}$$

Specifically, $\pi_\theta(y|z, x)$ indicates the probability of generating the target sequence $y$ given both the chain of thought $z$ and the input $x$, while $\pi_\theta(y|x)$ denotes the probability of generating $y$ with only the input $x$. With respect to Equation 1, the language model's initial belief about the value of actions or outputs is described by $\pi_\theta(y \mid x)$ (from Equation 3), whereas its final belief after generating a sequence of computations or tokens (a chain of thought $z$) is described by $\pi_\theta(y \mid z, x)$. The cost is directly proportional to the logarithm of the number of tokens in the chain of thought $l(z)$:

$$\mathcal{C}(z) = \gamma \cdot \log l(z). \tag{4}$$

The hyperparameter $\gamma$ scales the cost and utility to the same magnitude. We found that both linear and logarithmic cost functions perform adequately in our training algorithm, but the latter was better at balancing utility and cost, especially with higher sequence-length variance. A key benefit of this reward function is that it is parameterized by the same weights $\theta$ as the generative policy $\pi_\theta$, eliminating the need for an external reward model. This allows for direct estimation of the utility of a reasoning chain using the policy itself.

## 3.2 CoT GENERATION

We begin with a pretrained language model $\pi_\theta$ and an initial dataset of problems $x$ along with their corresponding correct final answers $y$: $\mathcal{D} = \{(x_i, y_i)\}_{i=1}^D$. Following prior work in online RL (Tang et al., 2024), we utilize the model itself to generate the reasoning chains, with few-shot prompting as a guide. Specifically, we concatenate a small set of examples, denoted as $\mathcal{P}$, each containing intermediate reasoning chains $z$, to each example in $\mathcal{D}$. For each task $\tau_i = (x_i, y_i)$ in the original dataset $\mathcal{D}$, we generate $K$ reasoning chains: $\hat{\tau}_i = \{(x_i, z_{k,i}, y_i)\}_{k=1}^K$. If none of the $K$ generated reasoning chains for a task $\tau_i$ leads to the correct answer, we discard all samples. To minimize the likelihood of this outcome, we adopt the rationalization approach introduced by STaR (Zelikman et al., 2022): if the model fails to generate the correct answer, we generate a new reasoning chain by supplying the model with the correct answer in the prompt. Intuitively, rationalization increases the sample size, providing more options when selecting the $\arg\max$ of the VoC reward. We found that it provided marginal reduction in length with no effect on performance. Finally, we assess each reasoning chain using the Rational Metareasoning reward function (Equation 2).

### 3.3 METAREASONING TRAINING

We demonstrate the effectiveness of our reward using a variation of the Expert Iteration algorithm (EI, Anthony et al., 2017). EI is known for its sample efficiency and strong performance on reasoning tasks (Havrilla et al., 2024; Zelikman et al., 2022). As an example of an online reinforcement learning (RL) algorithm, EI involves both exploration and policy improvement phases, with the policy $\pi_\theta$ being updated using data from the exploration phase. STaR (Zelikman et al., 2022) applies EI to LLMs by iteratively generating reasoning chains via few-shot prompting and compiling them into a fine-tuning dataset to refine the model. We build on this framework for our rational metareasoning training process, described in Algorithm 1.

---

**Algorithm 1** Rational Metareasoning Training

---

    **Input** $\pi$: a pretrained LLM; dataset $\mathcal{D} = \{(x_i, y_i)\}_{i=1}^{D}$

1:  $\pi_0 \leftarrow \pi$                                                  ▷ Copy the original model
2:  $\mathcal{D}_0 \leftarrow \emptyset$
3:  **for** $n$ **in** $1...N$ **do**                                        ▷ Iterations
4:     $\mathcal{D}_n \leftarrow \{\mathcal{T} \subseteq \mathcal{D} \mid |\mathcal{T}| = T\} \cup \mathcal{D}_{n-1}$            ▷ Sample batch from dataset
5:     **for** $k$ **in** $1...K$ **do**                     ▷ Perform reasoning chain generation
6:         $(z_{i,k}, y_{i,k}) \leftarrow \pi_{n-1}(x_i) \quad \forall i \in [1, D_i]$
7:         $(z_{i,k}, y_{i,k}) \leftarrow \pi_{n-1}(\text{add\_hint}(x_i)) \quad \forall i \in [1, D_i] \wedge y_{i,k} \neq y_i$   ▷ Compute rationalization
8:     **end for**
9:     $r_{i,k} \leftarrow \mathcal{R}_{\pi_{n-1}}(x_i, y_i, z_{i,k}) \quad \forall i, k(i \in [1, D_i] \wedge y_{i,k} = y_i)$     ▷ Compute reward
10:    $\hat{z}_i \leftarrow \arg\max_k \{r_{i,k}\}_{k=1}^{K}$            ▷ Select best reasoning chain for each task $i$
11:    $\mathcal{D}_n^* \leftarrow \{(x_i, \hat{z}_i, y_i) \in [1, D_i]\}$          ▷ Create the optimal dataset
12:    $\pi_n \leftarrow \text{train}(\pi, \mathcal{D}_n^*)$       ▷ Finetune the original model on the optimal solutions
13: **end for**

---

Initially, in the exploration phase, we approximate the optimal policy $\hat{\pi}^*$ (whose computations maximize the VoC reward) by using rejection sampling on our student policy $\pi_\theta$. After generating $K$ intermediate reasoning chains $z_1, \ldots, z_K$ for a given question $x$ (Section 3.2), we evaluate them using our reward function $\mathcal{R}_\pi$ (Section 3.1). We then construct $\mathcal{D}_1^* = \{(x_i, z_i, y_i)\}_{i=1}^{N}$ by setting the rejection sampling threshold dynamically to the highest reward for each task $i$, therefore increasing robustness to hyperparameter choices : $z_i = \arg\max_k \{\mathcal{R}_\pi(x, z_k, y)\}_{k=1}^{K}$. These rollouts are distilled into a policy $\pi_1$ using standard cross-entropy loss. This process can be iteratively repeated to refine the policy $\pi_n$ on the dataset $\mathcal{D}_n^*$. Finally, similar to STaR (Zelikman et al., 2024), instead of utilizing the entire training dataset at each iteration, as is typical in standard EI algorithms (Anthony et al., 2017; Gulcehre et al., 2023), we begin with a batch of $T$ steps during the first iteration and progressively increase the number of fine-tuning training steps by $T$ with each subsequent iteration. This approach allows the model to encounter new examples gradually, resulting in slower training initially, which ultimately enhances model performance.

## 4 EXPERIMENTS

We now detail the datasets (Sec. 4.1), baselines (Sec. 4.2), and training (Sec 4.3) used to evaluate our method.

### 4.1 DATASETS

In our efforts to develop a general-purpose reasoning model, we applied our training process and assessed its effectiveness across a diverse range of datasets and reasoning tasks. We constructed our training set by combining the training sets from these datasets into one dataset $\mathcal{D}$ and then evaluated the model on all corresponding test sets $\mathcal{T}$. Below is a detailed description of the datasets used:

- **ARC** (Clark et al., 2018). The AI2 Reasoning Challenge (ARC) dataset comprises grade-school science questions, designed to evaluate a model's capability to apply scientific knowledge.

- **CommonsenseQA** (Talmor et al., 2019). This dataset is centered on commonsense question answering. It leverages implicit human knowledge that is commonly known and sensible, testing the model's ability to provide answers based on everyday reasoning.

- **GSM8K** (Cobbe et al., 2021). This dataset includes a variety of linguistically diverse grade school math word problems. It assesses the model's proficiency in solving mathematical problems that require comprehension and application of arithmetic reasoning.

- **ProofWriter** (Tafjord et al., 2021). This dataset assesses logical deductive reasoning by asking the model to determine if a conclusion can be inferred from premises presented in natural language.

These datasets have very different train split sizes. To ensure fairness and balance between the datasets, and to manage computational costs, we composed our training mixture by sampling 1,024 random samples from each of the training sets. We then evaluated the model on the public test set of each dataset. To further assess the generalization of our approach, we conducted out-of-distribution testing on **MMLU** (Hendrycks et al., 2021), a massive multitask benchmark consisting of multiple-choice questions from various branches of knowledge. Again, to limit computational costs, we conducted the test on the first 100 samples of each subject within the MMLU benchmark, giving us a total of 5700 questions across 57 subjects.

## 4.2 BASELINES

We illustrate the advantages of our model by comparing its performance to two types of prompting strategies: **Direct prompting**, where the model is required to provide an immediate answer, and **Chain of Thought prompting (CoT)**, where the model is encouraged to reason through the problem step-by-step before arriving at a solution. Since we are using pretrained models which are not specifically trained for instruction following, we provide five few-shot examples for each task from the unused portion of the training dataset. These examples are carefully chosen to ensure that the length of the reasoning chain matches the perceived difficulty of the question. In addition to these prompting methods, we adopt a finetuning baseline, comparing our method to **STaR** (Self-Taught Reasoner; Zelikman et al. 2022), which also uses the expert iteration algorithm. While CoT prompting may not yield optimal trajectories, more advanced methods (Yao et al., 2023a; Zheng et al., 2024; Madaan et al., 2023) often increase sequence lengths. Focusing on efficiency, we find CoT's simplicity ideal for testing our method. Reducing reasoning tokens without performance loss here suggests similar gains in more complex approaches.

## 4.3 TRAINING DETAILS

For our experiments, we use Microsoft Phi-2 (Javaheripi et al., 2023), Meta Llama-3-8B (Dubey et al., 2024), and Mistral-7B-v0.3 (Jiang et al., 2023) as the pretrained base models. We have chosen $\gamma = 0.1$ to align the distributions of costs and rewards more closely, although we have found the method to be robust to small variations in the choice of this hyperparameter. We sample $K = 4$ reasoning chains for each question, using a temperature $t$ of 0.5 and a $top_p$ value of 0.9. These parameters are chosen to balance the exploration and exploitation trade-off, allowing us to generate diverse yet relevant reasoning chains. For each iteration $n$, we sample a dataset $\mathcal{D}_n$ of size 512 from the union of four training datasets $\mathcal{D}$ described in 4.1. In the self-supervised fine-tuning step, we use a batch size of 16 and a learning rate of 1e-6. We believe that further improvements are possible through a more comprehensive hyperparameter search; however, due to computational constraints, we leave this for future work. Finally, we evaluate all models using greedy decoding to ensure consistent and deterministic output generation. We use pattern matching techniques to extract the answers; an exact match between the generated answer and the ground truth is considered correct.

| Model | Method | Accuracy (%) ↑ | # Input Tokens ↓ | # Output Tokens ↓ |
|---|---|---|---|---|
| phi-2 | Direct Few-Shot | 50.7 (± 1.3) | 441.6 (± 0.9) | 0.0 (± 0.0) |
| | CoT Few-Shot | 56.6 (± 1.3) | 1047.3 (± 0.9) | 193.9 (± 4.3) |
| | STaR | **63.2** (± 1.5) | **75.1** (± 0.9) | 156.9 (± 2.4) |
| | Metareasoning | **64.7** (± 1.4) | **75.1** (± 0.9) | **123.1** (± 2.1) |
| Meta-Llama-3-8B | Direct Few-Shot | 55.5 (± 1.3) | 421.3 (± 0.9) | 0.0 (± 0.0) |
| | CoT Few-Shot | **63.7** (± 1.4) | 995.0 (± 0.9) | 148.4 (± 3.1) |
| | STaR | **63.9** (± 1.3) | **73.3** (± 0.8) | 119.7 (± 2.0) |
| | Metareasoning | **64.4** (± 1.4) | **73.3** (± 0.9) | **95.6** (± 2.2) |
| Mistral-7B-v0.3 | Direct Few-Shot | 52.7 (± 1.4) | 498.2 (± 1.0) | 0.0 (± 0.0) |
| | CoT Few-Shot | 57.5 (± 1.5) | 1180.9 (± 1.1) | 182.7 (± 4.3) |
| | STaR | **60.1** (± 1.3) | **83.2** (± 0.9) | 157.4 (± 2.0) |
| | Metareasoning | **60.5** (± 1.5) | **83.2** (± 1.0) | **114.4** (± 2.2) |

Table 1: Comparison of different methods based on accuracy and length metrics, averaged across datasets (means with 95% confidence intervals; bold indicates best performing approaches with overlapping 95% intervals; see Tables 4 and 5 for per-dataset results). Metareasoning achieves comparable performance while using significantly fewer input and output tokens compared to STaR or CoT Few-Shot prompting.

## 5 RESULTS

### 5.1 PERFORMANCE VS COST

We first evaluate our approach against baselines (Sec. 4.2) across several datasets (Sec. 4.1). Our key criteria are *performance* (measured by accuracy) and *cost* (measured by the number of input and output tokens). Our experiments confirm that across all three models and four datasets, our training approach reduces cost while matching or improving performance (see Table 1 for results averaged across datasets; Tables 4 and 5 for per-dataset results; and Appendix E for example reasoning chains). Fig. 1 shows these results for all three models: the y axis compares cost across different baselines and datasets, while the x axis compares performance. We first consider the performance of our baselines: CoT Few-Shot prompting uses a large number of input and output tokens, but yields reasonable performance. In contrast, Direct Few-Shot prompting uses fewer input (and far fewer output) tokens, but yields poor performance on reasoning-intensive datasets (GSM8K and Proofwriter, see Table 4). STaR improves on all of these approaches, using significantly fewer tokens, while achieving comparable or superior performance. Finally, Metareasoning further improves the cost-performance tradeoff by matching STaR's accuracy while generating 20-37% fewer tokens on average.

### 5.2 ADAPTIVE COMPUTATION

Section 5.1 demonstrates that our method reduces computational costs *on average*. But does it actually teach models to reason *adaptively* (by adjusting reasoning to match task complexity), or just to reason *less*? To address this question, we first divided our test set $\mathcal{T}$ based on whether or not reasoning was needed to obtain the correct answer. We split the data based on whether Direct Few-Shot on Phi-2 obtained the correct answer. This yielded a split of 4412 "easy" and 2700 "hard" examples. Adaptive methods should use less computation to solve the easy problems. We can empirically compare the results across methods for these two data splits. As shown in Table 2, all models and methods are able to differentiate between hard and easy problems, generating fewer tokens on easier problems.[1] Intriguingly, STaR *decreases* the difference

---

[1] We note that we specifically chose CoT few shot examples that demonstrated adaptive reasoning length; we expect the effects here to be dependent on the chosen examples.

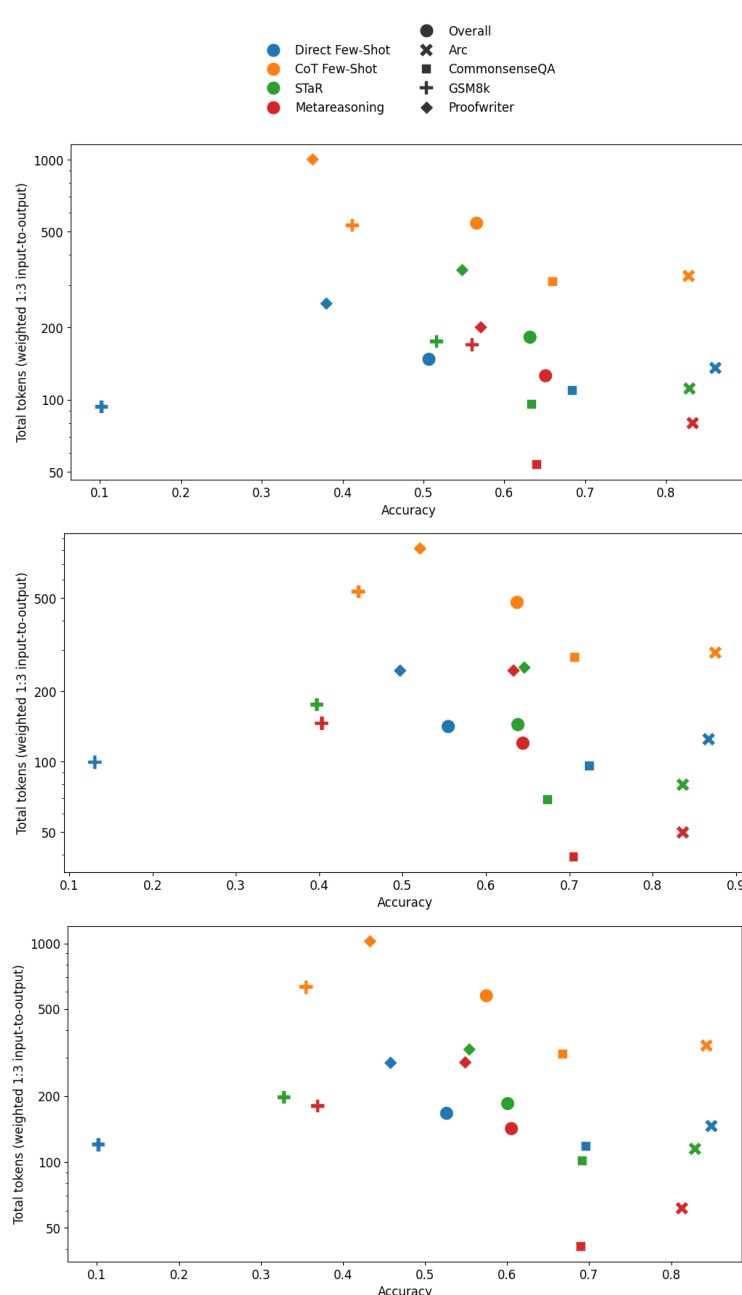

Figure 1: **Cost and performance**. The plot shows accuracy against the logarithm of the weighted sum of input and output tokens (weighted 3:1, as is common in API pricing).

| Model | Method | Hard Split | Easy Split | Length Reduction (%) ↑ |
|---|---|---|---|---|
| | CoT Few-Shot | 145.0 ($\pm$ 4.0) | 86.0 ($\pm$ 2.0) | 40.7 |
| Phi-2 | STaR | 135.0 ($\pm$ 4.0) | 91.0 ($\pm$ 1.0) | 32.6 |
| | Metareasoning | 122.0 ($\pm$ 4.0) | 57.0 ($\pm$ 3.0) | **53.3** |
| | CoT Few-Shot | 132.0 ($\pm$ 3.0) | 57.0 ($\pm$ 1.0) | 56.8 |
| Meta-Llama-3-8B | STaR | 126.0 ($\pm$ 3.0) | 64.0 ($\pm$ 2.0) | 49.2 |
| | Metareasoning | 99.5 ($\pm$ 2.5) | 27.0 ($\pm$ 1.0) | **72.9** |
| | CoT Few-Shot | 144.0 ($\pm$ 4.0) | 77.0 ($\pm$ 3.0) | 46.5 |
| Mistral-7B-v0.3 | STaR | 149.0 ($\pm$ 6.0) | 98.0 ($\pm$ 1.0) | 34.2 |
| | Metareasoning | 125.0 ($\pm$ 3.0) | 32.0 ($\pm$ 1.0) | **74.4** |

Table 2: The length of generated reasoning chains across models and methods (mean number of tokens with 95% confidence intervals). Adaptive methods should maximize the difference in length between the two distributions. Our method (Metareasoning) reduces the overall length and increases the difference in the length distribution between the Hard and Easy splits (as seen in column Length Reduction), demonstrating an improvement in the model's ability to adapt the reasoning length to the complexity of the task.

| Model | Method | Accuracy (%) ↑ | # Output Tokens ↓ |
|---|---|---|---|
| Phi-2 | STaR | 47.9 ($\pm$ 1.3) | 110.5 ($\pm$ 1.5) |
| | Metareasoning | 48.1 ($\pm$ 1.3) | **78.7** ($\pm$ 1.5) |
| Meta-Llama-3-8B | STaR | 56.3 ($\pm$ 1.1) | 75.4 ($\pm$ 1.4) |
| | Metareasoning | 54.5 ($\pm$ 1.4) | **48.8** ($\pm$ 2.1) |
| Mistral-7B-v0.3 | STaR | 54.5 ($\pm$ 1.3) | 112.0 ($\pm$ 1.6) |
| | Metareasoning | 54.0 ($\pm$ 1.4) | **50.6** ($\pm$ 1.5) |

Table 3: Comparison of different methods based on accuracy and length metrics in an out-of-distribution setting on the MMLU benchmark. We report the mean with 95% confidence intervals. Metareasoning achieves comparable performance while generating significantly fewer output tokens compared to STaR.

in reasoning length between hard and easy problems, suggesting that it trains out this adaptive tendency, biasing the model towards using reasoning on all problems. In contrast, Rational Metareasoning *increases* the difference in reasoning between hard and easy problems, achieving a length reduction of up to 74.4% on Mistral 7B. This indicates that our approach trains models to reason adaptively, helping them recognize when detailed reasoning is necessary and when a shorter response is sufficient.

## 5.3 GENERALIZATION

We assess the out-of-distribution generalization of our method using the **MMLU** benchmark (Hendrycks et al., 2021, see Section 4.1). As presented in Table 3, our approach achieves comparable performance while generating 28.8% to 54.8% fewer tokens compared to STaR. Among the 57 subcategories of MMLU, Fig. 2 highlights four—high school chemistry, mathematics, U.S. history, and macroeconomics—that effectively showcase the strengths of our model. We specifically chose two subjects (U.S. history and macroeconomics) where the use of chain-of-thought (CoT) reasoning appears to be counterproductive compared to Direct Few-Shot prompting, as well as two subjects (mathematics and chemistry) where the benefits of CoT are more pronounced. Consistent with our findings from the four in-domain datasets, we observe that the reduction in output length achieved by our method is inversely related to the benefits gained from intermediate reasoning

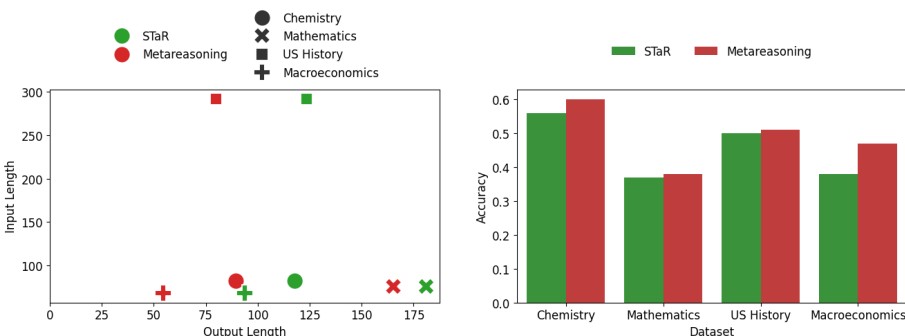

Figure 2: **Cost and Performance for Phi-2 Across Selected MMLU Domains**. Left panel: Input and output token distributions for different methods and domains. Right panel: While CoT improves performance in subjects like mathematics and chemistry, it appears less effective in history and macroeconomics. The latter domains show a greater reduction in output tokens (35% for history and 37% for macroeconomics), compared to the smaller reductions seen in chemistry (20%) and mathematics (12%), where intermediate reasoning chains provide more substantial benefits.

chains. In the cases of U.S. history and macroeconomics, our method results in a substantial reduction in generated tokens. In contrast, in mathematics and chemistry, the reduction in token count is smaller.

# 6 RELATED WORK

## 6.1 REDUCING INFERENCE COSTS

The rising cost of deploying large language models (LLMs) has driven efforts to reduce inference costs. Techniques such as Speculative Decoding (Leviathan et al., 2023) and Medusa (Cai et al., 2024) improve efficiency through parallelization, while Mixture of Experts (Jacobs et al., 1991; Zhou et al., 2022) activates only a subset of LLM parameters during decoding. Though effective, these methods require significant architectural changes and don't adapt computation based on task difficulty. Other approaches have developed neural architectures that enable adaptive computation (Graves, 2017; Banino et al., 2021; Dehghani et al., 2019; Mohtashami et al., 2023; Schuster et al., 2022) but involve new architectures or training methods. In contrast, our approach uses existing architectures and pretrained models, modifying only the fine-tuning process. More similar to our approach, model routing (Ong et al., 2024; Jiang et al., 2024) optimizes resource utilization based on query complexity by routing easier queries to smaller models and harder queries to larger ones. However, this necessitates multiple models and a router, while our approach trains a single model to adaptively adjust its own outputs to match task complexity.

## 6.2 REASONING IN LLMS

Techniques such as Chain of Thought (CoT) and related methodologies (Wei et al., 2023; Yao et al., 2023a; Yasunaga et al., 2024; Madaan et al., 2023; Zheng et al., 2024) have proven effective at enhancing LLM performance across a wide range of tasks. CoT boosts LLMs' performance on complex reasoning by guiding them through a series of intermediate reasoning steps, increasing inference costs to improve task performance. This method can be implemented through in-context learning (Wei et al., 2023), prompting (Kojima et al., 2023), or training (Li et al., 2023). The benefits of CoT can be attributed to both a greater computation depth (Goyal et al., 2024; Pfau et al., 2024) and the semantic values of the thought tokens, which function as

intermediate variables in the computation of the answer (Prystawski et al., 2023). However, recent studies have raised concerns regarding the meaningfulness of such reasoning chains in reaching the target solution, and whether models effectively utilize them to solve tasks (Turpin et al., 2023; Paul et al., 2024; Sprague et al., 2024). We further demonstrate that standard prompting and training methods fail to teach the model to use CoT purposefully, resulting in inefficient inference. Reasoning can also be used to bootstrap language models. Self-improving techniques (Huang et al., 2022; Zelikman et al., 2022) consist of generating reasoning chain-augmented answers for unlabeled questions and fine-tuning the LLM using those self-generated solutions as target outputs. Similar techniques, such as ReST (Gulcehre et al., 2023), can also be used to better align LLMs with human preferences and needs. Our approach builds on these techniques to optimize the inference cost of reasoning chains in addition to their task performance. Another related work, Quiet-STaR (Zelikman et al., 2024), has improved LLM reasoning performance by modifying pretraining to generate intermediate s between tokens. While effective for downstream tasks, it increases computational costs by generating reasoning chains at every step, even when unnecessary. More recently, chat models that "think before answering" have been developed (OpenAI, 2024), using inference-time computation to enhance their outputs. Although these models outperform others, they expend more computational resources on all tasks, even when it may not be necessary. Our method could be incorporated into their training process to help the model determine when this additional computation is genuinely beneficial.

## 7 CONCLUSION

In this work, we introduced a novel method inspired by previous research in AI and cognitive science aimed at reducing inference costs in LLMs. We operationalize the concept of rational metareasoning – which formalizes humans' adaptive use of cognitive resources – with a novel reward function based on the Value of Computation (VOC). This reward function trains the LLM to optimize the use of intermediate reasoning steps in task execution, enabling it to balance task performance with computational efficiency. Empirically, we find that this approach significantly reduces the number of generated tokens and input context length while maintaining comparable performance across diverse datasets.

However, we note some limitations to our approach. First, the focus of our work here is enhancing the efficiency of large language models in reasoning tasks, rather than the overall performance of such systems. Thus, we aimed to reduce unnecessary intermediate reasoning steps, rather than improve the quality of reasoning per se. It remains to be seen whether our approach can be extended to enhance task performance in addition to computational efficiency. Additionally, we tested our approach on well-established datasets in science, commonsense reasoning, and math, but it remains untested in more specialized domains. Notably, one of these untested settings is the agentic one, where LLMs act as agents to perform complex tasks in digital environments (Yao et al., 2023b; Schick et al., 2023). Our method could be adapted by including the number of LLM API calls to such tools in the cost section of our reward function, ideally encouraging the model to minimize unnecessary tool calls. Broader testing could help assess its generalizability and effectiveness across diverse contexts. Additionally, the VoC reward function is highly adaptable and can be used with various algorithms. For instance, it can provide numerical rewards for PPO or guide DPO by training on samples with the highest and lowest rewards. In this work, we focused on EI, the most sample-efficient algorithm for reasoning tasks Havrilla et al. (2024), to demonstrate its generalization across models and benchmarks. Exploring its application to other algorithms is an exciting direction for future research.

Most excitingly, our work demonstrates how cognitively-inspired reward functions can endow LLMs with desirable inference-time properties, opening a broad avenue of future work. Given its flexibility, this method could be integrated into instruction tuning to potentially enhance performance, even in scenarios where verifying the correctness of answers is challenging. Since the utility measure within the reward function can be tailored to prioritize any desired, measurable property, this approach offers the potential to guide models toward achieving these enhanced qualities while still benefiting from the reduced computational costs.

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

## A  FULL RESULTS TABLES

| Model | Method | ARC | Commonsenseqa | GSM8K | Proofwriter |
|---|---|---|---|---|---|
| Phi-2 | Direct Few Shot | 86.1 (± 1.1) | 68.4 (± 2.6) | 10.2 (± 1.6) | 38.0 (± 3.0) |
| | Cot Few Shot | 82.8 (± 1.2) | 66.0 (± 2.6) | 41.2 (± 2.8) | 36.3 (± 3.1) |
| | STaR | 82.9 (± 1.1) | 63.4 (± 2.9) | 51.6 (± 2.8) | 54.8 (± 2.8) |
| | Metareasoning | 83.9 (± 1.3) | 61.6 (± 2.7) | 54.9 (± 2.6) | 58.2 (± 2.9) |
| Meta-Llama-3-8B | Direct Few Shot | 86.7 (± 1.0) | 72.4 (± 2.5) | 13.1 (± 1.8) | 49.7 (± 3.1) |
| | Cot Few Shot | 87.5 (± 1.2) | 70.6 (± 2.5) | 44.7 (± 2.6) | 52.1 (± 3.0) |
| | STaR | 83.6 (± 1.2) | 67.4 (± 2.7) | 39.7 (± 2.5) | 64.6 (± 2.7) |
| | Metareasoning | 83.6 (± 1.2) | 70.5 (± 2.7) | 40.3 (± 2.8) | 63.3 (± 2.8) |
| Mistral-7B-v0.3 | Direct Few Shot | 84.9 (± 1.3) | 69.6 (± 2.6) | 10.2 (± 1.6) | 45.8 (± 2.9) |
| | Cot Few Shot | 84.3 (± 1.2) | 66.8 (± 2.7) | 35.5 (± 2.4) | 43.3 (± 2.9) |
| | STaR | 82.9 (± 1.4) | 69.2 (± 2.5) | 32.8 (± 2.4) | 55.4 (± 3.0) |
| | Metareasoning | 81.3 (± 1.2) | 69.0 (± 2.3) | 36.9 (± 2.8) | 54.9 (± 3.2) |

Table 4: Comparison of accuracy of different methods across datasets and models. We report the mean with 95% confidence scores.

| Model | Method | ARC | Commonsenseqa | GSM8K | Proofwriter |
|---|---|---|---|---|---|
| Phi-2 | Direct Few Shot | 0.0 (± 0.0) | 0.0 (± 0.0) | 0.0 (± 0.0) | 0.0 (± 0.0) |
| | Cot Few Shot | 74.6 (± 1.4) | 80.4 (± 2.2) | 161.0 (± 2.9) | 459.6 (± 16.5) |
| | STaR | 91.4 (± 1.0) | 80.6 (± 1.5) | 155.7 (± 3.1) | 299.9 (± 7.2) |
| | Metareasoning | 55.2 (± 1.2) | 53.0 (± 1.7) | 143.8 (± 2.6) | 240.4 (± 6.8) |
| Meta-Llama-3-8B | Direct Few Shot | 0.0 (± 0.0) | 0.0 (± 0.0) | 0.0 (± 0.0) | 0.0 (± 0.0) |
| | Cot Few Shot | 59.5 (± 0.9) | 69.9 (± 2.0) | 157.1 (± 2.8) | 307.1 (± 11.4) |
| | STaR | 61.3 (± 1.2) | 55.7 (± 1.9) | 155.3 (± 2.8) | 206.4 (± 6.2) |
| | Metareasoning | 31.4 (± 0.7) | 26.1 (± 0.7) | 125.8 (± 4.7) | 199.2 (± 8.2) |
| Mistral-7B-v0.3 | Direct Few Shot | 0.0 (± 0.0) | 0.0 (± 0.0) | 0.0 (± 0.0) | 0.0 (± 0.0) |
| | Cot Few Shot | 71.0 (± 1.4) | 66.2 (± 2.3) | 174.8 (± 3.6) | 418.8 (± 16.9) |
| | STaR | 93.1 (± 1.3) | 85.0 (± 1.9) | 175.7 (± 3.8) | 275.9 (± 5.7) |
| | Metareasoning | 39.7 (± 1.0) | 25.0 (± 0.7) | 158.6 (± 3.6) | 234.3 (± 7.2) |

Table 5: Comparison of mean output length of different methods across datasets and models. We report the mean with 95% confidence scores.

## B  ALTERNATIVE PLOT OF ACCURACY VS COST

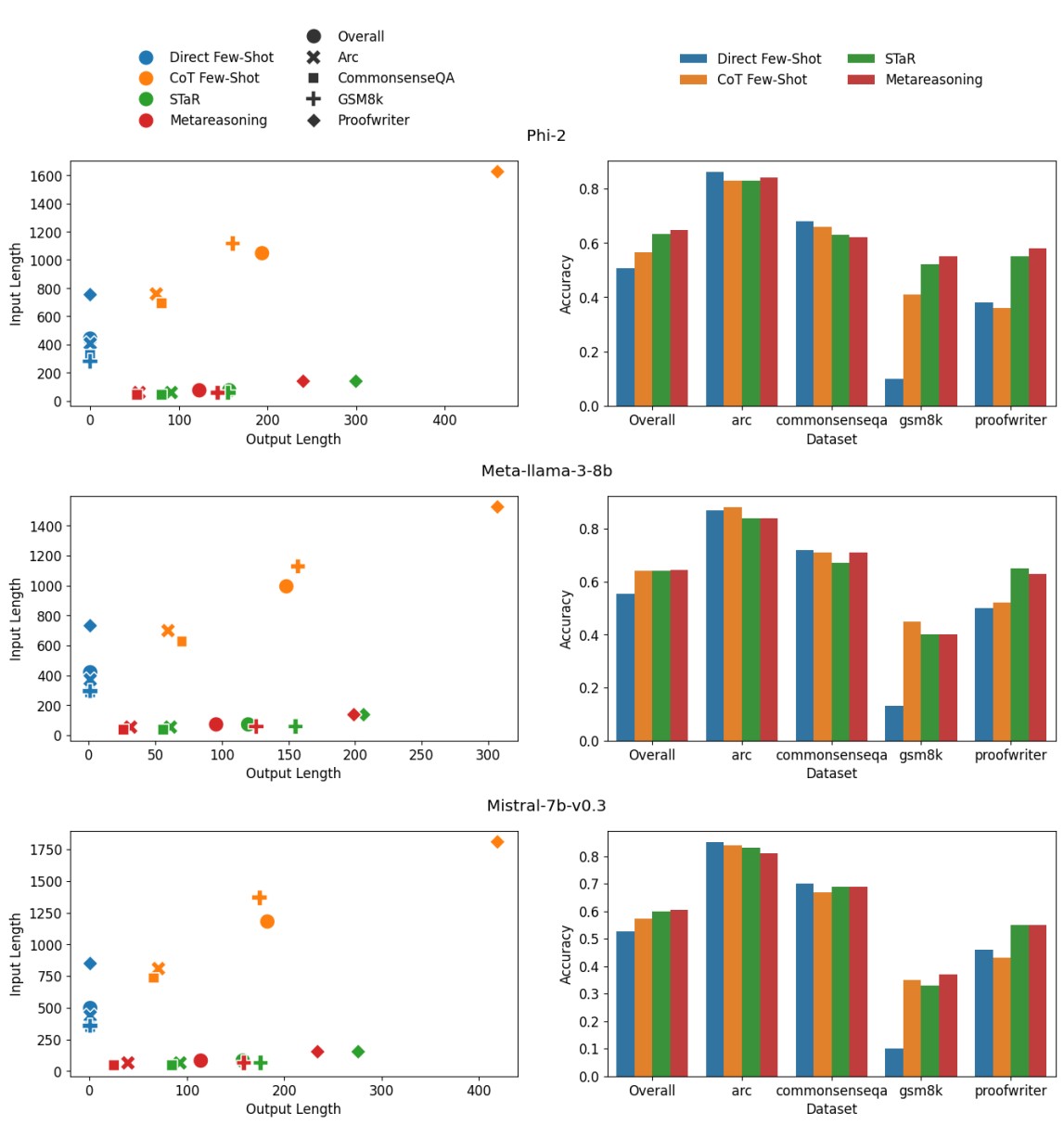

Figure 3: **Cost and performance**. Left panel: the distribution of input and output tokens across various methods and datasets. Our method, Metareasoning, eliminates the need for few-shot prompting (reducing input tokens) and trains the model to use fewer reasoning tokens compared to STaR. Right panel: Metareasoning matches or improves accuracy relative to other methods, despite substantially reducing inference costs. Across datasets, the reduction in token count (left) is inversely related to the effectiveness of the CoT (right): the model learns that CoT is essential in GSM8K and Proofwriter, but less necessary in CommonsenseQA.

| Dataset | Method | Accuracy (%) ↑ | # Input Tokens | # Output Tokens |
|---------|--------|----------------|----------------|-----------------|
| MMLU | Direct Few-Shot | 34.1 (± 1.1) | 495.7 (± 2.1) | 0.0 (± 0.0) |
|  | CoT Few-Shot | 50.7 (± 1.3) | 1050.7 (± 2.2) | 99.1 (± 2.2) |
|  | STaR | 48.2 (± 1.3) | 94.7 (± 2.1) | 110.5 (± 1.5) |
|  | Metareasoning | 48.3 (± 1.4) | 94.7 (± 2.2) | 78.7 (± 1.6) |
| Aqua-Rat | Direct Few-Shot | 22.4 (± 5.1) | 477.9 (± 2.5) | 0.0 (± 0.0) |
|  | CoT Few-Shot | 38.6 (± 5.5) | 1032.9 (± 2.6) | 152.6 (± 11.3) |
|  | STaR | 42.9 (± 5.9) | 76.9 (± 2.5) | 138.8 (± 11.5) |
|  | Metareasoning | 36.6 (± 5.9) | 76.9 (± 2.4) | 111.2 (± 7.7) |

Table 6: **Additional OOD testing on Phi-2.** This table compares the performance of our method against STaR, CoT and Direct Few-Shot across two out-of-domain benchmarks: MMLU and AQUA-RAT. The results highlight that our method achieves comparable accuracy while requiring fewer output tokens, demonstrating both efficiency and robustness in diverse reasoning tasks.

## C  ADDITIONAL OUT OF DOMAIN TESTING

To further evaluate the out-of-domain generalization capabilities of our method, we conducted additional assessments on AQUA-RAT (Ling et al., 2017) and MMLU Hendrycks et al. (2021) using the Phi-2 model. For MMLU, we incorporated a few-shot prompting baseline with examples drawn from datasets used during training and compared the results to STaR and Metareasoning. AQUA-RAT was tested under the same conditions. The results indicate that our method performs comparably to few-shot prompting and STaR on both datasets, while requiring fewer output tokens. This analysis highlights the strength of our approach in achieving a balance between computational efficiency and accuracy in OOD scenarios. Comprehensive results for MMLU and AQUA-RAT are provided in Table 6.

## D  COMPARISON TO INSTRUCTION TUNED MODELS

It is reasonable to question whether instruction-tuned models can dynamically adjust the length of their reasoning chains based on task complexity. To explore this, we tested Meta-Llama-3-8B-Instruct using the following prompt:

```
Answer the following question, thinking step by step to get to the
answer.  You can think however long you need, but answer as soon as
you're ready.  Keep you response concise and use the minimum number of
steps to get to the answer.  Once you're finished thinking, write your
answer after the 'Answer:  '  prompt.
```

Unfortunately, this method did not effectively adapt response lengths to align with task complexity in our experiments, as shown in Tables 7 and 8 and Fig. 4. While the observed performance drop might partially result from deviations from the expected input format, it is evident that the length of the reasoning chains was generally much higher and not meaningfully adaptive to the complexity of the task.

## E  QUALITATIVE EXAMPLES

Below we show one example of question, reasoning chain and answer per dataset, generated with each method: CoT Few Shots (Yellow), STaR (green) and Metareasoning (Red). The following examples were generated by Phi-2 (Javaheripi et al., 2023). The reasoning chains shown here serve as intermediate computations

| Model | Method | Accuracy (%) ↑ | # Input Tokens | # Output Tokens |
|---|---|---|---|---|
| | Direct Few-Shot | 55.5 ($\pm$ 1.2) | 421.3 ($\pm$ 1.0) | 1.0 ($\pm$ 0.0) |
| | CoT Few-Shot | 63.7 ($\pm$ 1.4) | 995.0 ($\pm$ 0.9) | 148.4 ($\pm$ 3.0) |
| Meta-Llama-3-8B | CoT Instruct | 57.7 ($\pm$ 1.3) | 73.3 ($\pm$ 0.9) | 190.4 ($\pm$ 2.5) |
| | STaR | 63.9 ($\pm$ 1.5) | 73.3 ($\pm$ 1.0) | 119.7 ($\pm$ 1.9) |
| | Metareasoning | 64.4 ($\pm$ 1.4) | 73.3 ($\pm$ 0.9) | 95.6 ($\pm$ 2.3) |

Table 7: **Llama Instruct results.** The table shows accuracy (in %), average number of input tokens, and average number of output tokens for each method. Despite the instruction, reasoning chains remain lengthy.

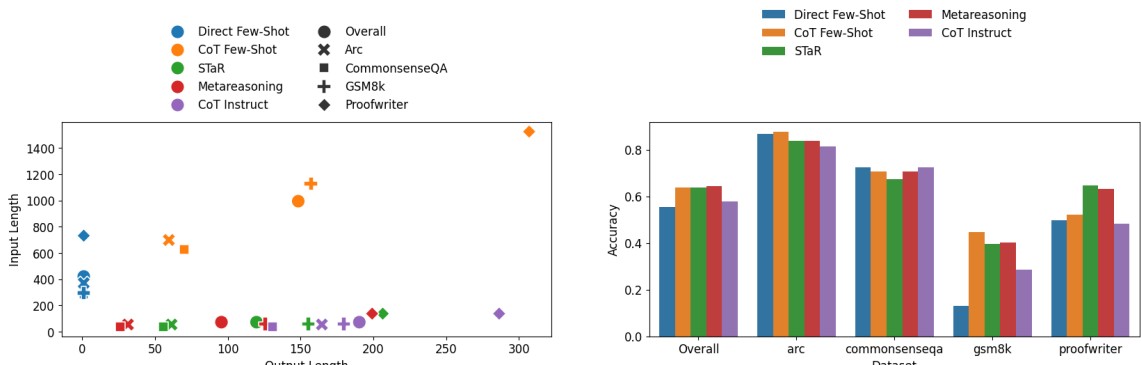

Figure 4: **Llama Instruct plot.** Left panel: the distribution of input and output tokens across various methods and datasets. Right panel: Comparison of performance across datasets.

| Model | Method | Hard Split | Easy Split | Length Reduction (%) ↑ |
|---|---|---|---|---|
| | CoT Few-Shot | 132.0 ($\pm$ 3.0) | 57.0 ($\pm$ 1.0) | 56.8 |
| Meta-Llama-3-8B | CoT Instruct | 181.0 ($\pm$ 4.0) | 150.0 ($\pm$ 2.0) | 17.1 |
| | STaR | 126.0 ($\pm$ 3.0) | 64.0 ($\pm$ 2.0) | 49.2 |
| | Metareasoning | 99.5 ($\pm$ 2.5) | 27.0 ($\pm$ 1.0) | **72.9** |

Table 8: The length of generated reasoning chains across models and methods (mean number of tokens with 95% confidence intervals). Adaptive methods should maximize the difference in length between the two distributions. Our method (Metareasoning) reduces the overall length and increases the difference in the length distribution between the Hard and Easy splits (as seen in column Length Reduction), demonstrating an improvement in the model's ability to adapt the reasoning length to the complexity of the task.

Which natural process is responsible for the formation of the Mississippi Delta?
(A) Snow
(B) Drought
(C) Deposition
(D) Weathering

**Cot Few-Shots**

I need to identify the natural process that leads to the formation of the Mississippi Delta.
1. Snow: Snow does not contribute to the formation of the Mississippi Delta.
2. Drought: Drought can lead to the drying up of rivers, but it does not directly cause the formation of the delta.
3. Deposition: Deposition is the process by which sediment is dropped or laid down, and it is the primary process responsible for the formation of the Mississippi Delta.
4. Weathering: Weathering is the breakdown of rocks and minerals, but it does not directly contribute to the formation of the delta.

**STaR**

I need to identify the natural process that is responsible for the formation of the Mississippi Delta.
1. Snow: The accumulation of snow on mountains.
2. Drought: A period of abnormally low rainfall.
3. Deposition: The process by which sediment is dropped or laid down.
4. Weathering: The breaking down of rocks and minerals.

**Metareasoning**

The Mississippi Delta is formed by the deposition of sediment carried by the Mississippi River.

Answer: (C) Deposition

Figure 5: Qualitative examples of reasoning processes for ARC dataset

that guide the model to its final answers. While not intended as direct explanations, the examples remain interpretable and demonstrate the effectiveness of the respective methods.

The artist was sitting quietly pondering, then suddenly he began to paint when what struck him?
(A) Sadness
(B) Anxiety
(C) Inspiration
(D) Discomfort
(E) Insights

The question asks what inspired the artist to start painting.
- Sadness: This is not a likely source of inspiration.
- Anxiety: This is not a likely source of inspiration.
- Inspiration: This is a likely source of inspiration.
- Discomfort: This is not a likely source of inspiration.
- Insights: This is not a likely source of inspiration.

I need to find the word that best completes the sentence in a way that makes logical sense. The sentence implies that the artist was thinking deeply about something before he began to paint.
The word that best completes the sentence in a way that makes logical sense is (C) Inspiration.

The question asks what struck the artist while he was sitting quietly pondering. The correct answer should be something that would inspire him to start painting.

Answer: (C) Inspiration

Figure 6: Qualitative examples of reasoning processes for CommonsenseQA dataset

Brandon's iPhone is four times as old as Ben's iPhone. Ben's iPhone is two times older than Suzy's iPhone. If Suzy's iPhone is 1 year old, how old is Brandon's iPhone?

**Cot Few-Shots**

I need to determine the age of Ben's iPhone first by multiplying Suzy's iPhone age by 2. Then, I'll multiply Ben's iPhone age by 4 to find Brandon's iPhone age.
1. Calculate Ben's iPhone age:
 Ben's iPhone = Suzy's iPhone * 2 = 1 * 2 = 2 years old
2. Calculate Brandon's iPhone age:
 Brandon's iPhone = Ben's iPhone * 4 = 2 * 4 = 8 years old

**STaR**

I need to determine the age of Ben's iPhone first, then use that information to find the age of Brandon's iPhone.
1. Determine the age of Ben's iPhone:
 Ben's_iPhone_Age = Suzy_iPhone_Age * 2 = 1 * 2 = 2 years
2. Determine the age of Brandon's iPhone:
 Brandon_iPhone_Age = Ben_iPhone_Age * 4 = 2 * 4 = 8 years

**Metareasoning**

I need to determine the age of Ben's iPhone and then find the age of Brandon's iPhone.
1. Determine the age of Ben's iPhone:
 Ben_Age = Suzy_Age * 2 = 1 * 2 = 2 years
2. Find the age of Brandon's iPhone:
 Brandon_Age = Ben_Age * 4 = 2 * 4 = 8 years

Answer: 8

Figure 7: Qualitative examples of reasoning processes for GSM8K dataset

**Premises:** Anne is furry. Bob is smart. Bob is white. Dave is quiet. Erin is rough. Erin is not smart. Erin is young. All rough people are furry. If someone is white and not young then they are cold. Smart people are rough.
**Conclusion:** Erin is not furry.
**Is the conclusion True, False or Unknown based on the premises?**

Let's analyze the premises step by step to determine if Erin is furry.
1. Anne is furry: IsFurry(Anne).
2. Bob is smart: IsSmart(Bob).
3. Bob is white: IsWhite(Bob).
4. Dave is quiet: IsQuiet(Dave).
5. Erin is rough: IsRough(Erin).
6. Erin is not smart: ~IsSmart(Erin).
7. Erin is young: IsYoung(Erin).
8. All rough people are furry: IsRough(x) -> IsFurry(x).
9. If someone is white and not young then they are cold: IsWhite(x) and ~IsYoung(x) -> IsCold(x).
10. Smart people are rough: IsSmart(x) -> IsRough(x).
From the premises, I know that Erin is rough: IsRough(Erin).Let's look at the rules:
1. All rough people are furry: IsRough(x) -> IsFurry(x). Since Erin is rough, Erin is furry: IsFurry(Erin).
2. If someone is white and not young then they are cold: IsWhite(x) and ~IsYoung(x) -> IsCold(x). Since Erin is not young, Erin is cold: IsCold(Erin).
3. Smart people are rough: IsSmart(x) -> IsRough(x). Since Erin is smart, Erin is rough: IsRough(Erin).
Therefore, the conclusion that Erin is not furry is False.

Let's analyze the premises step by step to determine if Erin is furry.
1. Anne is furry: IsFurry(Anne).
2. Bob is smart: IsSmart(Bob).
3. Bob is white: IsWhite(Bob).
4. Dave is quiet: IsQuiet(Dave).
5. Erin is rough: IsRough(Erin).
6. Erin is not smart: ~IsSmart(Erin).
7. Erin is young: IsYoung(Erin).
8. All rough people are furry: IsRough(x) -> IsFurry(x). Since Erin is rough, Erin is furry: IsFurry(Erin).
Hence, the conclusion that Erin is not furry is False.

Let's analyze the premises step by step to determine if Erin is furry.
1. Erin is rough: IsRough(Erin).
2. All rough people are furry: IsRough(x) -> IsFurry(x). Since Erin is rough, Erin is furry: IsFurry(Erin).
Hence, the conclusion that Erin is not furry is False.

**Answer: False**

Figure 8: Qualitative examples of reasoning processes for Proofwriter dataset

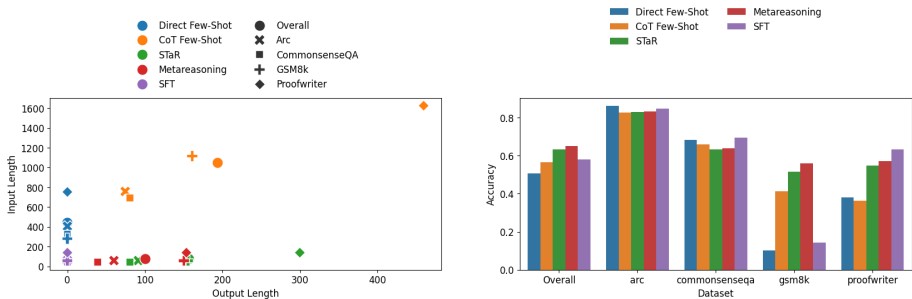

Figure 9:

## F    DIRECT FINETUNING BASELINE

We also tested our method against a direct finetuning baseline. It can be seen that while this method provides an even bigger efficiency gain, the performance is less consistent.

