# OpenReview forum: "Rational Metareasoning for Large Language Models"
_ICLR.cc/2025/Conference — Submitted to ICLR 2025_

### Official Review · Reviewer_mBzh · 2024-10-16

**Soundness:** 3
**Presentation:** 3
**Contribution:** 2
**Rating:** 5
**Confidence:** 5

**Summary:**

This paper presents an approach for optimizing small LLMs towards generating shorter chains of reasoning while maintaining capabilities. To this end, the authors designed a reward function that balances the log-likelihood of generating a target answer with the log normalized cost of the number of generating chains of thought tokens. Experiments on a number of question-answering datasets demonstrate that the proposed approach can reduce the number of output tokens while maintaining the LLM’s performance.

**Strengths:**

A) The paper is clear and easy to follow. Furthermore, the method is simple enough and the paper detailed enough so that it would, I believe, be easy to reproduce.

B) The problem of reducing the number of output tokens of CoT-like approaches has practical relevance.

C) While the LLMs that were investigated are on the small end (2.7B-8B), the authors demonstrate that their approach leads to reduced number of output tokens on multiple LLMs and datasets.

D) I appreciate the inclusion of insightful qualitative results in the Appendix.

**Weaknesses:**

## Major
E) I appreciate the attempt to demonstrate the validity of the method for multiple chains of reasoning approaches (CoT and STaR). However, in my view, the paper is incremental, since the cost of generating chains of reasoning has already been addressed by follow up work like Zelikman et al. (2024). Quiet-STaR: Language Models Can Teach Themselves to Think Before Speaking. arXiv. https://doi.org/10.48550/arXiv.2403.09629 The authors should directly compare to Quiet-STaR, and investigate whether their approach can provide orthogonal gains.

F) In my view, the following important ablation is missing to ensure that the models are indeed retaining their ability to reason (and decompose a problem into subproblem) as opposed to relying simply on directly producing an answer. Looking at Figure 3 and Figure 4 in the Appendix, it seems to me the model learned to simply rely on its internalized knowledge (one conclusion from this could be that ARC and CommonsenseQA simply aren’t great benchmarks to assess complex reasoning capabilities). As a baseline, directly fine-tune the LLM to produce the correct answer. I also need to point out that the reduction in reasoning lengths is much less impressive on the harder reasoning benchmarks GSM8K and Proofwriter (see Table 5). Thus, my current hypothesis is that the proposed approach works well for problems that are not hard reasoning problems, but is much less effective on harder reasoning problems where we actually need more CoT-like methods.

G) Related to the above, I believe the authors should test on additional hard compositional problems such as AQUA-RAT (https://github.com/google-deepmind/AQuA).

H) Another ablation I am missing is investigating to what extent RL is really needed here. What happens if you generate chains of reasoning on the training data, ask the LLM to summarize these chains, and then directly fine-tune on this synthetically generated data?

## Minor
I) LLMs are on the smaller scale, so it remains to be seen whether gains in terms of reduced number of output tokens persist for larger models.

**Questions:**

I don’t have any questions, the paper was very clear.

What the authors would have to demonstrate to see an improved rating from me:
1. Compare directly to Quiet-STaR, and demonstrate whether their approach leads to orthogonal gains (E)
2. Add an ablation where the LLM is directly fine-tuned on the author’s training sets (F)
3. Demonstrate results on additional compositional reasoning benchmarks such as AQUA-RAT since ARC and CommonsenseQA aren’t informative (G)
4. Add an ablation where the LLM is used to summarize chains of reasoning and then fine-tuned on the resulting synthetic data (H)

---

> ### Author Response · Authors · 2024-11-22
>
> Thank you for your comments. We hope to address your concerns and clarify the purpose of our paper in the following response:
>
> ---
>
> ### (E)
> Quiet-STaR has indeed improved LLM reasoning performance by modifying pretraining to generate intermediate reasoning chains between tokens. While this is effective for downstream tasks, it increases computational costs by generating reasoning chains at every step, even when unnecessary. In contrast, our approach is specifically designed to enhance *efficiency*. Additionally, Quiet-STaR operates at the continual pretraining scale, whereas our approach is applied at the fine-tuning scale, making the methodologies distinct.
>
> We appreciate the suggestion to explore combining these methods, but we were unable to do so within the timeline of the rebuttal. It would be interesting to investigate whether our VoC reward could similarly reduce the cost of Quiet-STaR, as we have demonstrated for standard CoT.
>
> ---
>
> ### (F)
>
> Thank you for flagging this! We agree with your observations, but we believe they actually motivate and support our method. We’ll try to clarify what we mean here, and will update the Results section accordingly.
>
> > Looking at Figure 3 and Figure 4 in the Appendix, it seems to me the model learned to simply rely on its internalized knowledge
>
> This is correct. One of the key objectives of our approach, Rational Metareasoning, is to train the model to only use reasoning when necessary. When the model already has the knowledge accessible, reasoning is superfluous, and RM-trained models use shorter reasoning chains. This is in line with our claims: RM reduces inference costs while maintaining performance compared to other reasoning methods, which cause the model to generate costly but often useless reasoning tokens.
>
> > one conclusion from this could be that ARC and CommonsenseQA simply aren’t great benchmarks to assess complex reasoning capabilities
>
> We agree with this as well! Again, our method is intended to improve efficiency rather than performance. We use these datasets to highlight the fact that standard reasoning methods are insufficiently flexible, causing them to waste tokens when reasoning is not needed.
>
> > I also need to point out that the reduction in reasoning lengths is much less impressive on the harder reasoning benchmarks GSM8K and Proofwriter (see Table 5). Thus, my current hypothesis is that the proposed approach works well for problems that are not hard reasoning problems, but is much less effective on harder reasoning problems where we actually need more CoT-like methods.
>
> You are again correct that we see a less significant reduction in token generation for benchmarks which do require reasoning (GSM8K and Proofwriter). However, we also view this as a strength of our method: we still reduce inference costs on these datasets, without compromising performance.
>
> The kind of adaptive use of different amounts of reasoning demonstrated across datasets is actually one of the strengths of Rational Metareasoning. The pattern of “significant reasoning reductions on ARC/CommonsenseQA and small reductions on GSM8K/Proofwriter” demonstrates that Rational Metareasoning is in fact teaching the LLM to spend inference time adaptively, based on what is effective in each task.
>
> We also added an the requested ablation where the model is directly finetuned on the answer for the phi-2 model, without intermediate CoT (*Appendix F*). It can be seen that while this provides an even bigger efficiency gain, the performance is less consistent.
>
> ---
>
> ### (G)
>
> Thank you for highlighting this relevant benchmark. Although we did not have the opportunity to include this dataset in the training set, we tested the out-of-domain generalization capabilities of the methods on its test set, as detailed in Appendix C. The results show that performance remains comparable, while our method uses significantly fewer reasoning tokens, confirming its efficiency.
>
> ---
>
> ### (H)
>
>  Thank you for your thoughtful suggestion. The models used in our study were not instruction-finetuned, so they lacked the ability to effectively summarize their reasoning. While your proposed approach is compelling, it differs from the central objectives of our work. Our primary focus is on enabling the model to autonomously determine when additional reasoning is required and to improve itself through a self-improvement loop. While fine-tuning on reasoning chains generated by bigger models or humans is an interesting idea that we agree should be explored, it does not align with the core objective of our work, which emphasizes use of reinforcement learning without involving external guidance.
>
> ---
>
> ### (I)
>
> Unfortunately, due to limited computational resources, we were not able to test our method on excessively large models.

---

> > ### Comment · Reviewer_mBzh · 2024-11-25
> >
> > Thank you for your rebuttal. I have improved my score accordingly.

---

### Official Review · Reviewer_hJGT · 2024-10-23

**Soundness:** 2
**Presentation:** 2
**Contribution:** 3
**Rating:** 5
**Confidence:** 4

**Summary:**

This paper proposes a new fine-tuning approach for training LLMs to generate token-efficient and adaptive responses. The method encourages the LLM to produce longer reasoning chains only when necessary. Building on the STaR framework, it introduces a filtering step to fine-tune the model on correct answers with the highest reward, where the reward function penalises unnecessarily long responses. Experiments on a dataset constructed from a mixture of four benchmark datasets show that the method effectively reduces the token count of generated responses without sacrificing accuracy. The results also demonstrate that the method promotes adaptive behaviour, producing longer responses for more difficult questions and shorter ones for simpler questions.

**Strengths:**

- The paper tackles an important and underexplored problem of lowering sampling costs at inference time without sacrificing model performance in an adaptive manner.
- The proposed method shows improved performance over a set of 3 baselines, including the STaR method which is a direct ablation of Rational Metareasoning.
- The method is evaluated on a good selection of benchmark datasets.

**Weaknesses:**

- The link between VOC and the latter presented approach of modified expert iteration is unclear. Sections 2 and 3 are not well connected together.
- Incremental contribution. The method is an extension of existing work (STaR). More justification or analysis of this approach's novelty would strengthen the paper (see question 2).
- There a few places where writing could be improved (see the writing suggestions below).
- There are a few inconsistencies or unclear statements (see the clarity comments below).
- A potential limitation, not mentioned in the paper, is that while shorter responses may be preferred from the computational costs point of view, they may not necessarily be more human friendly. The paper would benefit from a small human study assessing the qualitative aspects of the generated reasoning chains with Rational Metareasoning.
- Experiments could include additional baselines and ablation studies (see below for details).


**Comments on clarity**

- Line 159: the statement: *“Initially, in the exploration phase, we approximate the optimal policy   by using rejection sampling on our student policy $\pi_\theta$”* is unclear.  First, the notion of an optimal policy $\hat{\pi}^*$ hasn’t been defined. Second, in rejection sampling, all reasoning chains with the reward above a certain threshold should be retained. Yet, the proposed algorithm only selects a single reasoning chain that maximises the reward. The motivation for this choice is unclear (see question 2).
- It seems that there are a few inconsistencies in the presentation of Algorithm 1. Assignment of $\pi_0$ is missing. In line 2, the entire dataset $\mathcal{D}$ is assigned to  $\mathcal{D}\_n$, but from the comment and the main text it follows that $\mathcal{D}\_n$ is subsampled from $\mathcal{D}$.  The reward function in the algorithm should should be made dependent on $\pi_{n-1}$, i.e.  $\mathcal{R}\_{\pi_{n-1}}$. Storing the rewards in $r_{i,k}$ (line 7 of Algorithm 1) seems redundant given the later $\arg\max$ again uses $\mathcal{R}$. The input to this argmax should be $\mathcal{R}\_{\pi_{n-1}}(x_i, z_{i,k}, y_i)$ rather than $\mathcal{R}(x, z_{i,k}, y)$, I believe. The quantifier $\forall i$ is also missing in this step.


**Experiments**

- It would be interesting to compare Metareasoning with Direct Few-Shot prompting, where the LLM is explicitly instructed to provide concise responses (e.g. “Keep your answer concise”.). It has been previously demonstrated that such a statement shortens the expected answer length, often without sacrificing the performance [1, 2].
- The batching technique with increasing $T$ is a design choice which should be tested in an additional ablation study.
- The proposed expert-iteration algorithm could be additionally compared to other fine-tunning algorithms, like PPO, to better motivate the particular choice of the training method.

**Minor writing suggestions**

- Line 032: “… many of these methods reduce costs at the expense of performance”—this statement would benefit by adding an example with an appropriate reference similar to the juxtaposed approach of chain-of-thought.
- Lines 071-075: the word “while” is repeated 4 times in 3 consecutive sentences, consider rephrasing.
- Line 097: There is no equation 3.1 in the paper

**References**

[1] https://arxiv.org/pdf/2401.05618

[2] https://arxiv.org/pdf/2407.19825

**Questions:**

1. Could the authors elaborate on how VOC translates to the setting of reasoning with LLMs? What would the computations $c$, beliefs $b$ and actions $a$ correspond to with respect to the terms presented in section 3: input $x$, reasoning chain $z$ and answer $y$?
2. Why is the LLM policy fine-tuned only with respect to the reasoning’s $\hat{z}$ that maximise the reward within the sampled batch of rasoning chains? In expert iteration, we would typically retain all reasoning chains $z_i$ for which the reward is above a certain threshold. Could the authors provide either a theoretical motivation behind their choice or run additional empirical studies comparing their training method to alternatives (e.g. expert iteration with threshold rejection, PPO) to justify the choice of the training method?
3. Is the $T$ in lines 165-169 referring to the number of fine-tunning steps used in the $\mathrm{train}(\pi, \mathcal{D}_n^*)$ subroutine at each iteration or is it the number of examples sampled from $\mathcal{D}$?
4. How many training iterations $N$ of the algorithm are needed for convergence and what is the relationship between the compute time for policy training vs. final performance?

---

> ### Author Response · Authors · 2024-11-22
>
> Thank you for your insightful comments. Regarding the potential limitation that shorter responses may not necessarily be human-friendly, we would like to clarify that we do not treat reasoning chains as explanations for the model's answers intended for humans. Instead, they serve as intermediate computations that assist the model in arriving at its answer. Nevertheless, we recognize the importance of maintaining interpretability. We have updated the discussion in the appendix *Appendix E* to address your concern.
>
> ---
>
> ### Comments on Clarity
>
> Thank you for catching these errors! We have corrected the algorithm and updated the paper (*main text, lines 166–167 and 169–170*) to address your valid concerns. Specifically, the optimal policy approximation refers to selecting reasoning chains that maximize the reward. These chains are then used to fine-tune the model, shifting its policy closer to the optimal one.
>
> ---
>
> ### Experiments
>
> 1. **Instruction-Tuned Model Baseline**
>    Thank you for this suggestion. While we did not test this baseline prior to submission, we agree it is an interesting approach (albeit applicable only to instruction-tuned models). We conducted an experiment using **Llama-3-8B-Instruct**. Unfortunately, this method did not effectively adapt response lengths to match task complexity in our setting. However, we acknowledge it as a noteworthy baseline, especially for larger models or with more refined prompt engineering. The paper now includes this baseline and discussion (*Appendix D*).
>
> 2. **Batching Technique**
>    This choice was partially based on the STaR approach, where the authors found that slower initial training benefits performance. In retrospect, we agree that an ablation study to test its impact on our method would be valuable. We have started further experiments to investigate this but may not complete them by the rebuttal deadline.
>
> ---
>
> ### Minor Writing Suggestions
>
> Thank you for these suggestions and corrections—we have updated the paper to address them!
>
> ---
>
> ### Questions
>
> 1. **Connection Between VoC and the Reward Function**
>    Upon re-reading the manuscript, we agree that this connection was not described in sufficient detail. First, individual reasoning tokens correspond to intermediate computations $c$, and the reasoning chain $z$ is a sequence of these computations. The actions $a$ represent the space of possible LLM outputs, while the target sequence $y$ is one such sequence ($y \in \mathcal{Y}$). The belief state $b$ corresponds to the LLM’s internal state after generating zero or more reasoning tokens, and the input $x$ determines the LLM’s initial belief state. Therefore, $\pi_\theta(y \mid x)$ (from Eq. 3) is equivalent to $\mathcal{E}[U(a) \mid b]$.
> However, an important distinction exists: Eq. 1 describes the value of an individual computation/token, whereas Eq. 2 describes the value of a sequence of computations (a reasoning chain). The LLM’s initial “belief” about the value of actions or outputs is captured by $\pi_\theta(y \mid x)$ (Eq. 3), and its final “belief” after reasoning is described by $\pi_\theta(y \mid z, x)$. We hope this explanation clarifies the connection; please let us know if further points are unclear. The paper has also been updated to clarify this connection (*main text, lines 109–111, 117–120*).
>
> 2. **Rejection Sampling Threshold and Alternative RL Algorithms**
>    A fixed threshold (e.g., 0) makes the hyperparameter $\gamma$ highly influential, as heavily weighting costs can yield negative rewards. We chose a dynamic threshold to increase robustness to hyperparameter choices.
>    Regarding alternative algorithms, the VoC reward function can be readily adapted to other algorithms (for example, DPO would entail training on samples with the highest and lowest reward respectively). We felt it was most important to demonstrate generalization across models and benchmarks, so in this work we focused on a single algorithm (expert iteration) because it has been found to be most sample efficient for reasoning [1]. Unfortunately, we did not have time or computational resources to experiment with alternatives during the rebuttal phase. However, we have updated the discussion (*main text, lines 459-463*) to clarify our motivation.
>
> 4. **T in Lines 165–169**
>    The $T$ refers to the number of examples sampled from $\mathcal{D}$.
>
> 4. **Expert Iteration Steps**
>    We perform up to 8 expert iteration steps, but performance typically plateaus after 4–5 iterations. Intermediate checkpoints were not preserved due to their size. We plan to rerun Phi-2 training with STaR and MR and will update the paper with these details once available.
>
> ### References
>
> [1] Alex Havrilla, Yuqing Du, Sharath Chandra Raparthy, Christoforos Nalmpantis, Jane Dwivedi-Yu, Maksym Zhuravinskyi, Eric Hambro, Sainbayar Sukhbaatar, and Roberta Raileanu. **Teaching large language models to reason with reinforcement learning**, 2024.

---

> > ### Comment · Reviewer_hJGT · 2024-11-24
> >
> > Thank you for your response which addressed my questions and concerns. I nevertheless decide to keep my score due to the incremental nature of the paper’s contributions.

---

### Official Review · Reviewer_6gdb · 2024-10-31

**Soundness:** 3
**Presentation:** 2
**Contribution:** 2
**Rating:** 5
**Confidence:** 4

**Summary:**

This paper presents a method for fine-tuning large language models to enhance the cost-performance trade-off in reasoning procedure. The authors introduce a reward function that evaluates the computational value of various reasoning chains, enabling the ranking of these chains. By integrating this reward function with expert Iteration, the method trains LLMs utilizing intermediate reasoning steps selectively, employing them according to the rewards. This approach significantly reduces inference costs on specific datasets while preserving task performance.

**Strengths:**

1. The authors introduce an interesting problem in LLM reasoning, optimizing LLMs’ inference cost and performance at the same time. This is important issue in using LLMs especially considering the LLMs inference cost is becoming larger.

2. The paper is well-written and easy to follow. Overall, I could follow the whole story that the authors want to present in this paper.

**Weaknesses:**

1. Lack experiments in more realistic datasets. LLMs are not limited to tasks in text space; they are frequently utilized as agents that interact with external tools to perform complex tasks in various environments. Incorporating experiments on more realistic datasets, such as GAIA [1] and ToolBench [2], would provide valuable insights into the model's performance in more complex reasoning scenarios.

2. Currently, the method and experiments focus exclusively on CoT reasoning, generating trajectories for model fine-tuning and serving as baselines. However, CoT may not always produce the optimal trajectory and should not be considered a strong baseline. It would be beneficial to explore alternative reasoning methods for fine-tuning the models. At the very least, a discussion or justification regarding the choice of reasoning methods should be included in the paper.

[1] Mialon, G., Fourrier, C., Swift, C., Wolf, T., LeCun, Y., & Scialom, T. (2023). Gaia: a benchmark for general ai assistants. arXiv preprint arXiv:2311.12983.
[2] Guo, Z., Cheng, S., Wang, H., Liang, S., Qin, Y., Li, P., ... & Liu, Y. (2024). StableToolBench: Towards Stable Large-Scale Benchmarking on Tool Learning of Large Language Models. arXiv preprint arXiv:2403.07714.

**Questions:**

N/A

---

> ### Author Response · Authors · 2024-11-22
>
> ### Weakness 1
>
> Thank you for raising this concern! We agree that agentic reasoning (e.g., ReAct [1] or Toolformer [2]) is an important and emerging domain, and our method may indeed be applicable in such settings. However, the task format for tool-based methods differs significantly, requiring tailored datasets for accurate evaluation.
>
> Additionally, measuring the computational cost of agentic behavior is challenging. Different methods produce varying numbers of LLM calls, and external tools introduce variable and often hard-to-quantify costs. For these reasons, we focused on CoT-style reasoning to clearly demonstrate the efficiency gains of our methodology.
>
> That said, we recognize that applying our method to agentic settings is a valuable direction for future work, which we neglected to mention in the original discussion. We have updated the paper (*main text, lines 454–458*) to include this point.
>
> ---
>
> ### Weakness 2
>
> We agree it was an oversight not to discuss other reasoning approaches in the paper. While we acknowledge that CoT prompting does not always produce the optimal trajectory, more complex methods, such as Tree of Thoughts [3], Step-back Prompting [4], and Self-Refinement [5], may achieve higher performance. However, our approach is designed to improve **efficiency**, not performance.
>
> CoT is the most basic and concise reasoning approach, making it an appropriate baseline for testing our method. The demonstrated ability of our method to reduce reasoning tokens without compromising performance in this minimal setting suggests that similar efficiency gains could extend to more complex methods, which typically involve significantly more reasoning steps.
>
> We hope this explanation clarifies our choice of CoT. The paper has been updated to reflect this motivation (*main text, lines 217–220*).
>
> ---
>
> ### References
>
> [1] Shunyu Yao, Jeffrey Zhao, Dian Yu, Nan Du, Izhak Shafran, Karthik Narasimhan, and Yuan Cao. **ReAct: Synergizing reasoning and acting in language models**, 2023. URL: [https://arxiv.org/abs/2210.03629](https://arxiv.org/abs/2210.03629).
>
> [2] Timo Schick, Jane Dwivedi-Yu, Roberto Dessì, Roberta Raileanu, Maria Lomeli, Luke Zettlemoyer, Nicola Cancedda, and Thomas Scialom. **Toolformer: Language models can teach themselves to use tools**, 2023. URL: [https://arxiv.org/abs/2302.04761](https://arxiv.org/abs/2302.04761).
>
> [3] Shunyu Yao, Dian Yu, Jeffrey Zhao, Izhak Shafran, Thomas L. Griffiths, Yuan Cao, and Karthik Narasimhan. **Tree of Thoughts: Deliberate problem solving with large language models**, 2023. URL: [https://arxiv.org/abs/2305.10601](https://arxiv.org/abs/2305.10601).
>
> [4] Huaixiu Steven Zheng, Swaroop Mishra, Xinyun Chen, Heng-Tze Cheng, Ed H. Chi, Quoc V Le, and Denny Zhou. **Take a Step Back: Evoking reasoning via abstraction in large language models**, 2024. URL: [https://arxiv.org/abs/2310.06117](https://arxiv.org/abs/2310.06117).
>
> [5] Aman Madaan, Niket Tandon, Prakhar Gupta, Skyler Hallinan, Luyu Gao, Sarah Wiegreffe, Uri Alon, Nouha Dziri, Shrimai Prabhumoye, Yiming Yang, Shashank Gupta, Bodhisattwa Prasad Majumder, Katherine Hermann, Sean Welleck, Amir Yazdanbakhsh, and Peter Clark. **Self-Refine: Iterative refinement with self-feedback**, 2023. URL: [https://arxiv.org/abs/2303.17651](https://arxiv.org/abs/2303.17651).

---

> > ### Comment · Reviewer_6gdb · 2024-11-30
> > **Thanks for the response**
> >
> > Thanks for the response. My first concern is properly addressed. While my second concern still holds. My question is whether your method could work well on other reasoning methods rather than challenging the performance of your method. More experiments should be added in revised version. I would like to keep my score.

---

### Official Review · Reviewer_Eudp · 2024-11-04

**Soundness:** 2
**Presentation:** 3
**Contribution:** 2
**Rating:** 5
**Confidence:** 3

**Summary:**

This paper introduces a rational metareasoning approach for LLMs, using a novel Value of Computation (VOC)-based reward function to reduce inference costs without compromising performance. Tested across diverse benchmarks, the method achieves significant token savings (20-37%) compared to standard prompting techniques. The results highlight this approach as an effective solution for cost-efficient reasoning in LLMs, though further evaluation on task variety would enhance the findings’ robustness.

**Strengths:**

* The paper introduces a unique rational metareasoning approach that balances inference cost with performance, addressing a crucial need in the efficient deployment of LLMs.
* The integration of the Value of Computation (VOC)-based reward function is well-designed and thoughtfully applied, showing careful consideration of LLM efficiency.
* The approach is tested across a diverse set of benchmarks, covering science knowledge, commonsense reasoning, math problem-solving, and logical deduction, as well as an out-of-domain generalization task (MMLU).
* Empirical results demonstrate notable reductions in token generation (20-37%), indicating that the method can achieve similar performance with fewer computational resources.
* The approach is valuable for scenarios where computational resources are limited, potentially benefiting applications needing cost-effective, high-quality language model outputs.

**Weaknesses:**

* Limited Analysis of Time Complexity: While the paper focuses on token reduction, it lacks an analysis of the time complexity of the proposed method. A deeper investigation into time savings would provide a clearer picture of its practical efficiency.
* Narrow Range of LLMs and Tasks: The method was primarily tested on a limited selection of benchmarks and model architectures. Broader experimentation across different LLMs and a wider range of tasks would strengthen claims about the generalizability of the approach.
* Limited Robustness Testing: Although the results are promising, the paper lacks robustness checks to assess how performance holds under varied conditions, such as noisy inputs or more complex task requirements.
* Scalability Concerns: The feasibility of scaling this method to very large models or highly complex tasks is not fully addressed, leaving open questions about its applicability in more computationally intensive scenarios.

**Questions:**

### Questions
* Section 3.2: What is the likelihood of not generating a correct answer? Will this happen frequently? Any comment or justification on this?
* Are the input tokens of failed cases counted as part of the input and output tokens?
* What's the time cost of the proposed method itself, rather than the inference costs? The complexity is largely from the online reinforcement learning used in the proposed methods, i.e., EI?
*  The reliance on a single, VOC-inspired reward function might limit flexibility across diverse task types. Is there any justification on using alternative reward structures could reveal greater adaptability and robustness?

### Minors
* Line 97: Eq. 3.1 --> Eq. (2)?

---

> ### Author Response · Authors · 2024-11-22
>
> ### Alternative Model Architectures
> Regarding the use of alternative model architectures, such as SSMs, we agree this would be an interesting direction for future research. In this work, however, we prioritized demonstrating generalization across widely used models and benchmarks.
>
> ### Robustness Testing on Noisy Inputs
> For robustness testing on noisy inputs, we hypothesize that performance would align with the baselines. Instead, we focused on testing in an out-of-domain setting, which we considered a higher priority. Unfortunately, due to limited computational resources, we were unable to test our method on excessively large models.
>
> ---
>
> ### Q1. Likelihood of Not Generating the Correct Answer
> We do not have specific statistics but estimate it to be similar to the model's performance in the few-shot prompting setting. Rationalization intuitively increases the sample size, offering more options when selecting the argmax of the VoC reward. We conducted a limited ablation experiment on the phi-2 model by removing the rationalization step. This resulted in a reduction of 12 tokens on average in the validation set but had no effect on performance. We have updated the paper to clarify the motivation behind using rationalization (*main text, lines 137–139*).
>
> ---
>
> ### Q2. Input Tokens in Failed Cases
> Yes, the input tokens of failed cases are included in the count of input and output tokens.
>
> ---
>
> ### Q3. Time Complexity Reduction
> The time complexity reduction is proportional to the reduction in tokens, as each token requires the same amount of time on the fixed-depth transformer architecture we used. However, this may vary with different architectures.
>
> ---
>
> ### Q4. Alternative Reward Functions
> This work focuses on exploring the potential of VoC when applied to LLMs. Investigating alternative reward functions is indeed an interesting direction for future research.

---

### Official Review · Reviewer_1h6D · 2024-11-06

**Soundness:** 3
**Presentation:** 1
**Contribution:** 2
**Rating:** 5
**Confidence:** 4

**Summary:**

The paper proposes to fine-tune language models with an objective that maximizes task performance while minimizing the (logarithm of) the number of tokens used. This objective is optimized with expert iteration. Results suggest that this objective does indeed help models use the extra reasoning tokens only when they're actually needed, thus reducing compute cost on easier tasks.

**Strengths:**

- Computational efficiency of LLM deployments is a timely topic relevant to the ICLR community.
- The proposed method is sufficiently simple that something like it might well get used in practice.
- The paper is well-written and clearly structured.

**Weaknesses:**

### a) Insufficient baselines and ablations -- I don't feel like I get the "shape" of the proposed method and the potential alternatives all that well.

A few notes here:

1. How necessary is rationalization (described in the paragraph on line 131)? I assume the authors only use it because it's also used in STaR, the main baseline? Relatedly (and importantly), STaR should be described in more detail in the paper, and justified as a relevant baseline -- I needed to open up the original STaR paper to remind myself of what it does.

2. Re design of the computational cost function: what happens when the cost is linear instead of logarithmic in the number of tokens? (Authors mention they tried linear cost but don't provide details.) How exactly is $\gamma$ chosen, and how sensitive is the method to this choice?

3. Did the authors try a prompting baseline such as asking the model to think however long it needs, but answer as soon as it's ready? I'd be interested in seeing this kind of baseline with a few prompt variations, as this can plausibly also help avoid spending many tokens on easy tasks & spend more tokens on harder tasks when needed -- especially in larger and more capable models. I think models might well already be tracking whether further reasoning is useful: such tracking could plausibly arise just from self-supervised pretraining on large corpora that include texts written by humans who're tracking this implicitly. (Efficiency of such a prompting baseline could be improved further with prompt distillation, but I don't think this would be important for this paper).

4. I'm also interested in variations of the proposed method with RLHF-style algorithms other than expert iteration (e.g. iterated DPO).

### b) I did not find Figures 1 and 2 particularly useful, especially the subplots on the left side. Also, is information from subplots on the right side duplicated in the tables?

1. One idea for improving the figures could be plotting task performance on the y axis and the computational cost (e.g. input + output tokens) on the x axis. It might then be clearer whether your method is on the pareto frontier, and generally help understand the performance-efficiency tradeoff.

2. Regarding Figure 2: I think here it'd be helpful to include 5-shot and/or (0-shot, CoT) benchmarks for MMLU, to get a sense of how STaR and your method compare to these.

**Questions:**

1. Re Section 4.3, how many expert iteration steps $n$ do you perform?

---

> ### Author Response · Authors · 2024-11-22
>
> ### A1. Rationalization and STaR Details
>
> Thank you for suggesting additional details on STaR. We agree that our original presentation was insufficient and have added the relevant details to the paper (*main text, lines 147–149*).
>
> For rationalization, we followed the approach described in the STaR paper. Intuitively, rationalization increases the sample size, providing more options when selecting the argmax of the VoC reward. For example:
>
> - **Without rationalization**: If there are 4 samples (1 correct, 3 incorrect), the correct sample might be chosen even if it has a suboptimal VoC due to its length.
> - **With rationalization**: Rationalization can increase the size of the sample set, potentially yielding 3 correct samples and 1 incorrect one. This provides more choices and allows us to select the best trade-off between utility and cost.
>
> However, we did not evaluate the impact of rationalization on the metrics used to assess the models prior to submission. Since receiving this feedback, we conducted a limited experiment on the phi-2 model, ablating the rationalization step. We found that rationalization provided a marginal reduction in length (12 tokens on average in the validation set) without affecting performance. The paper has been updated to include the motivation behind using rationalization (*main text, lines 137–139*).
>
> ---
>
> ### A2. Cost Function and Gamma Parameter
>
> We found that using a linear cost function resulted in slightly lower performance during early iterations on the validation set. A logarithmic cost function aligned utility and cost more effectively, especially given token-length variance.  However, your comment prompted us to conduct further testing, which revealed that the performance difference was negligible.
>
> Gamma, treated as a hyperparameter, was tested with values 1, 0.1, and 0.01 on the phi-2 model. Gamma = 0.1 provided the best balance, aligning cost and reward distributions effectively. Overall, the method is robust to variations in gamma, provided that we filter for correct answers (within 0.5 standard deviation in accuracy and five tokens in length).
>
> We have updated the paper (*main text, lines 118–120 and 250–252*) to reflect these choices and findings.
>
> ---
>
> ### A3. Instruction-Tuned Model Baseline
>
> Thank you for this suggestion. While we did not test this baseline prior to submission, we agree it is an interesting approach (albeit applicable only to instruction-tuned models). We conducted an experiment using **Llama-3-8B-Instruct**. Unfortunately, this method did not effectively adapt response lengths to match task complexity in our setting. However, we acknowledge that it is a noteworthy baseline, particularly as it may perform better with larger models, or with extensive prompt engineering. We have updated the paper to include this baseline and a corresponding discussion (*Appendix D*).
>
> ---
>
> ### A4. Exploring Other Algorithms
>
> We agree that this is an interesting open question! Rational metareasoning can be readily adapted to other algorithms (for instance, DPO could involve training on samples with the highest and lowest rewards, respectively). In this work, we prioritized demonstrating generalization across models and benchmarks. Consequently, we focused on a single algorithm (expert iteration) because it has been found to be the most sample-efficient for reasoning tasks.
>
> Unfortunately, due to time and computational constraints, we were unable to experiment with alternative algorithms during the rebuttal phase. However, we have updated the discussion (*main text, lines 459–463*) to clarify our rationale for choosing expert iteration and to outline directions for future work.
>
> ---
>
> ### B1. Visualization of Performance-Efficiency Tradeoff
>
> We’re sorry to hear that these figures were not helpful. We plotted accuracy against the logarithm of the weighted sum of input and output tokens (weighted 3:1, as is common in API pricing). We have included the plots in *Appendix B* of the paper. We hope this provides a clearer visualization of the performance-efficiency tradeoff.
>
> ---
>
> ### B2. Out-of-Domain Few-Shot Evaluation
>
> Thank you for this suggestion! Since this section focuses on testing out-of-domain generalization, we evaluated results using out-of-domain few-shot examples, specifically with 5 shots from the datasets used for training. We found that performance remained comparable, while our method used fewer output tokens. We have updated the paper (*Appendix C*) to include this information.
>
> ---
>
> ### Q1. Expert Iteration Steps
>
> We perform up to **8 expert iteration steps**. However, we observe that performance generally plateaus after iterations 4 or 5. Unfortunately, the intermediate checkpoints and their associated details were not preserved due to their large size. We plan to rerun the training of Phi-2 with STaR and MR and will update the paper with the corresponding training details once they are available.

---

> > ### Comment · Reviewer_1h6D · 2024-11-24
> >
> > Thank you for the thoughtful response! I appreciate the new figure (point B1) and the Llama-3-8B-Instruct baseline.
> >
> > Regarding the new appendix figure: I think the main text would benefit from some version of that, perhaps with slightly tweaked axes -- currently it's a bit unintuitive with the lower right corner being the best one (highest performance for lowest cost).
> >
> > Regarding the baseline: the prompt you show in the appendix seems fine, but of course it would be nice to see experiments with more capable models. How do you conclude that the model does not vary its response length based on task difficulty? Presumably you need to show smth similar to your Table 2? An alternate way of looking at this, by just assuming that different tasks are differently difficult, shows that Llama-3-8B-Instruct does adapt -- in your new Figure 4 (left), the ordering of how many tokens are used for the different tasks is the same for metareasoning and CoT instruct: CommonsenseQA < Arc < GSM8k < Proofwriter.

---

> ### Author Response · Authors · 2024-11-24
>
> ### Figure in Main Paper
>
> Thank you for your feedback. The choice to skew the y-axis was intentional for two reasons: first, we are optimizing for the logarithm of sequence length; second, we found that this representation enhances readability. Regarding including this figure in the main paper, we agree it would be a valuable addition. Being constrained with page limit, we decided to replace the main paper figure with this new one, and moved the old main figure to the appendix.
>
> ---
>
> ### Length Adaptation for Instruction-Tuned Models
>
> Thank you for pointing this out. We realized that we had omitted the table with the length reduction ratio and have now updated the Appendix to include it. As shown, the instruction-tuned version generates longer reasoning chains compared to the other methods, and is is less adaptive to task difficulty (length reduction ratio). While the order of the datasets is the same, we feel that the ratio between datasets and hard/easy questions is more indicative of the adaptive nature of the model.
>
> ---
>
> Please let us know if you have any further questions, we remain at your disposal.

---

> > ### Comment · Reviewer_1h6D · 2024-11-24
> >
> > Thanks for the response!
> >
> > Re figure, I have no issues with y axis logarithm; I meant to say that the figure would be more readable if the desired combination of token cost & accuracy was either top right or bottom left. Or perhaps if the y axis was labeled on the right side of the plot? BTW that figure should also include the new CoT baseline. Are some tables in the text duplicating information from the figures? You could make some space space by cutting duplicate information.
> >
> > Overall my main concern about the paper remains -- I don't feel like I get the "shape" of the proposed method and the potential alternatives all that well. Hence I am maintaining my score for the moment.

---

### Meta-Review · Area_Chair_tvCK · 2024-12-20

**Metareview:**

The paper proposes a fine-tuning approach to improve the token cost of LLMs on reasoning tasks. The approach is motivated by metareasoning, and trains a reward model to capture the value of computation when decoding tokens for reasoning tasks. Compared to Chain-of-Thought prompting and STaR as baselines, the proposed fine-tuning achieves significant reduction in tokens used without harming overall reasoning performance on some benchmarks.

All of the reviewers agreed that the problem is well-motivated and timely, and the solution approach is sound. However all of the reviewers concurred that the paper is borderline at present, and identified the following avenues for improving the experiments and the contributions:
1. Does VoC finetuning help improve token efficiency on other approaches to reasoning beyond CoT? (e.g. Quiet-STaR, other prompting approaches like Tree-of-Thought).
2. Ablations to identify the contributions from the different components of the proposed approach (e.g. is RL with the VoC reward model necessary, or could we accomplish similar results by distilling successful reasoning chains directly).
3. Experiments on more domains (e.g. AQuA) to ensure that the proposed approach is indeed adaptive to the hardness of tasks and domains.

**Additional Comments On Reviewer Discussion:**

The authors clarified some of the reviewers' confusions during the rebuttal (e.g. hyper-parameter choices) and studied the ability of instruction-tuned models to adapt the length of their responses through prompting (as a baseline compared to their more sophisticated VoC reward approach). The authors also tested their fine-tuned model out-of-distribution on the test set of one of the benchmarks suggested by reviewers, and found that there were token savings without compromising reasoning quality.
These clarifications helped improve the exposition of the paper. Addressing the remaining weaknesses (primarily the experimental evidence to establish that this approach can work across many domains, and against many baselines) will substantially strengthen the revised version of the paper.

---

### Decision · Program_Chairs · 2025-01-22

Reject